# Exploring the Wound Healing Potential of a *Cuscuta chinensis* Extract-Loaded Nanoemulsion-Based Gel

**DOI:** 10.3390/pharmaceutics16050573

**Published:** 2024-04-23

**Authors:** Nichcha Nitthikan, Weeraya Preedalikit, Kanittapon Supadej, Siripat Chaichit, Pimporn Leelapornpisid, Kanokwan Kiattisin

**Affiliations:** 1Department of Pharmaceutical Sciences, Faculty of Pharmacy, Chiang Mai University, Chiang Mai 50200, Thailand; nichcha_n@cmu.ac.th (N.N.); siripat.chaichit@cmu.ac.th (S.C.); pimporn.lee@cmu.ac.th (P.L.); 2Department of Cosmetic Sciences, School of Pharmaceutical Sciences, University of Phayao, Phayao 56000, Thailand; weeraya.pr@up.ac.th; 3Department of Medical Technology, School of Allied Health Sciences, University of Phayao, Phayao 56000, Thailand; kanittapon.su@up.ac.th; 4Innovation Center for Holistic Health, Nutraceuticals, and Cosmeceuticals, Faculty of Pharmacy, Chiang Mai University, Chiang Mai 50200, Thailand

**Keywords:** *Cuscuta chinensis*, nanoemulsions, wound care, anti-inflammation, molecular docking

## Abstract

*Cuscuta chinensis* (*C. chinensis*) presents many pharmacological activities, including antidiabetic effects, and antioxidant, anti-inflammatory, and antitumor properties. However, the wound care properties of this plant have not yet been reported. Therefore, this research aimed to evaluate the antioxidant, anti-inflammatory, and antibacterial activities of ethanol and ethyl acetate *C. chinensis* extracts. The phytochemical markers in the extracts were analyzed using high-performance liquid chromatography (HPLC). Then, the selected *C. chinensis* extract was developed into a nanoemulsion-based gel for wound care testing in rats. The results showed that both of the *C. chinensis* extracts exhibited antioxidant activity when tested using 2,2-Diphenyl-1-picrylhydrazyl (DPPH), ferric reducing antioxidant power (FRAP), and lipid peroxidation inhibition assays. They reduced the expression of IL-1β, IL-6, and TNF-α in RAW264.7 cells induced with lipopolysaccharide (LPS). The ethyl acetate extract also had antibacterial properties. Kaempferol was found in both extracts, whereas hyperoside was found only in the ethanol extract. These compounds were found to be related to the biological activities of the extracts, confirmed via molecular docking. The *C. chinensis* extract-loaded nanoemulsions had a small particle size, a narrow polydispersity index (PDI), and good stability. Furthermore, the *C. chinensis* extract-loaded nanoemulsion-based gel had a positive effect on wound healing, presenting a better percentage wound contraction Fucidin cream. In conclusion, this formulation has the potential for use as an alternative wound treatment and warrants further study in clinical trials.

## 1. Introduction

Non-communicable diseases (NCDs) are noninfectious health conditions that are often caused by lifestyle and environmental factors, such as an unhealthy diet, alcohol consumption, smoking, and a lack of exercise [1]. NCDs, such as hypertension, heart disease, stroke, cancer, and diabetes mellitus, affect the public health system in Thailand. Diabetes mellitus (DM) is a group of metabolic diseases characterized by chronic hyperglycemia, resulting from defects in insulin secretion, insulin action, or both [2]. A critical problem in diabetes patients is not only hyperglycemia, but also the complications of diabetes caused by peripheral arterial disease or peripheral neuropathy. These complications lead to poor ulcer healing, infection, and even leg amputation. Diabetic foot problems, including acute ulcers, chronic ulcers, and the requirement of amputation, have a major impact on the health-related quality of life of patients with diabetes [3]. Moreover, patients with complications have to pay higher costs because of their illness.

The use of medicinal plants for the treatment of various skin conditions has been popular for decades. Some natural medicines derived from plant extracts can be used for treating many diseases, including skin problems and wounds. *C. chinensis* Lam. (Dodder or Foi Thong in Thai) is a parasitic plant that wraps around other plants and uses them for its nourishment. This plant has no leaves, and its stems are thin, twining, filiform, glabrous, yellowish or pale yellowish, and approximately 1 mm in diameter. The seeds are 1–2 mm long, pale brown, and not smooth. *C. chinensis* has been traditionally used as a tonic and aphrodisiac in Thailand and other Asian countries [4]. A number of scientific publications have reviewed *C. chinensis*’s stems and seeds and have found that they have many pharmacological activities, i.e., antidiabetic effects, sexual function improvement, anti-tyrosinase activity and melanogenesis inhibition, hepatoprotective activity, anti-inflammatory activity in Alzheimer’s patients, renoprotective effects, anti-osteoporotic activity, and antitumor activity [5,6,7,8]. A previous study showed that *C. chinensis* could inhibit inflammation and help to heal chronic wounds. This study found that quercetin in the *C. chinensis* seed extract could decrease the production of nitrite in macrophages, which is related to an anti-inflammatory effect and the proliferation of unusual cells [9]. In addition, kaempferol and calycopteretin found in the extract had the potential to inhibit lipid oxidation and scavenge free radicals. *C. chinensis* seed water extract was found to protect murine osteoblastic MC3T3-E1 cells against tertiary butyl hydroperoxide (TBHP)-induced injury. It also inhibited reactive oxygen species generation, inhibited malondialdehyde (MDA) production, and increased superoxide dismutase activity [10]. A methanol extract of *C. chinensis* seeds was able to reduce inflammatory cytokines, tumor necrosis factor-α (TNF-α), interleukin-1β (IL-1β), interleukin-6 (IL-6), nuclear factor-κB (NF-κB), cyclooxygenase, MDA, and nitric oxide in the edema tissues of λ-carrageenan-induced paw edema mice [11]. Moreover, it was found that *C. chinensis* seed extract could suppress the phosphorylation of extracellular-signal-regulated kinase 1/2, Jun N-terminal kinase, p38 mitogen-activated protein kinase, and NF-κB p65 in activated microglia [12]. Another study found that *C. chinensis* extract exhibited therapeutic potential for the treatment of hind-limb ischemia, as it promoted peripheral angiogenic and anti-inflammatory effects in mice [13].

Nanoemulsions are dispersing systems that consist of an oil phase, a water phase, and an emulsifier. They have small droplet sizes, ranging from 100 to 600 nm, with a transparent or translucent appearance. The small droplet size provides stability against flocculation, creaming, and coalescence. Nanoemulsions can be used as delivery systems for both water-soluble and water-insoluble substances. They can be prepared using the following two methods: the high-energy emulsification method and the low-energy emulsification method [14]. Many previous studies have used extracts or active compounds loaded in nanoemulsions or nanoemulsion-based gels for wound healing, such as curcumin-loaded fish scale collagen hydroxypropyl methyl cellulose nanogels, *Elaeis guineensis* Jacq. leaf extract nanoemulsions, thymoquinone-loaded topical nanoemulgels, alginate-chitosan hydrogel patches with beta-glucan nanoemulsions, and curcumin nanoemulgels [15,16,17,18]. Nanoemulsions have many advantages for wound care. For example, they protect the drug from hydrolysis and oxidation, allow for controlled drug delivery to the wound site and rapid absorption, easily penetrate into the skin layer, and accelerate the wound healing process [19]. Therefore, nanoemulsions are appropriate formulations for overcoming the inherent limitations associated with traditional dosage forms, such as poor solubility, skin permeation, and stability issues, thereby maximizing the therapeutic potential of extracts for wound healing.

Therefore, this research aimed to evaluate the biological activities, including antioxidant, antimicrobial, and anti-inflammatory activities, and the toxicity of *C. chinensis* seed extracts. Additionally, an effective *C. chinensis* seed extract-loaded nanoemulsion-based gel was developed for wound care in an animal model. This study added value to the use of underutilized plants as an alternative treatment in wound care.

## 2. Materials and Methods

### 2.1. Materials

#### 2.1.1. Plant Material

*C. chinensis* seed powder was purchased from Shanghai Huibo International Trade Co., Ltd. (Wuhan, China).

#### 2.1.2. Microorganisms

The test microorganisms, namely, *Enterococcus faecalis* (DMST 2860), *Pseudomonas aeruginosa* (DMST 4739), and *Escherichia coli* (DMST 4212), were purchased from the Department of Medical Sciences, Ministry of Public Health, Bangkok, Thailand (DMST).

#### 2.1.3. Cell Line and Cytokines

The RAW264.7 cell was purchased from PromoCell, Germany. The PCR primers for β-actin, IL-1β, IL-6, and TNF-α were purchased from Eurofins MWG Operon, Konstanz, Germany.

#### 2.1.4. Chemicals

The Tris base, glacial acetic acid, and EDTA were purchased from Ajax, Australia. Dulbecco’s modified Eagle medium (DMEM), fetal bovine serum (FBS), and 3-(4,5-dimethylthiazol-2-yl)-2,5-diphenyltetrazolium bromide (MTT) were purchased from Invitrogen, Paisley, UK. The lipopolysaccharides (LPS) were purchased from Qiagen, Hilden, Germany. The 2,2-Diphenyl-1-picrylhydrazyl (DPPH), ferrous chloride (FeCl_2_), ferric chloride (FeCl_3_), ferrous sulfate (FeSO_4_), 2,2′-Azobis(2-methylpropionamidine) dihydrochloride (AAPH), kaempferol, hyperoside, Transcutol^®^ HP (diethylene glycol monoethyl ether), Cremophor^®^ RH40 (PEG-40 hydrogenated castor oil), linoleic acid, and phosphotungstic acid were purchased from Sigma–Aldrich, Steinheim, Germany. The 2,4,6-Tris(2-pyridyl)-s-triazine (TPTZ) was purchased from FlukaBuchs, Buchs, Switzerland. The hexane, ethyl acetate, ethanol, methanol, acetonitrile, acetate buffer, hydrochloric, dimethyl sulfoxide (DMSO), and deionized water were purchased from RCI Lab-scan, Bangkok, Thailand. The ammonium thiocyanate (NH_4_SCN) and phosphate buffer were purchased from Loba Chemie™, Mumbai, India. The sodium lauryl sulfate (SLS), Tween 80 (Polysorbate 80), polyglyceryl-3 polyricinoleate (PGPR), avocado oil, argan oil, and grape seed oil were purchased from Chanjao longevity, Bangkok, Thailand. Normal saline solution was purchased from Klean and Kare, Bangkok, Thailand.

### 2.2. Methods

#### 2.2.1. Plant Extraction

*C. chinensis* seed powder was extracted following the method described in our previous study [20]. The *C. chinensis* seed powder was macerated with ethyl acetate at a ratio of 1:5 g/mL for 48 h (3 cycles). The powder was then subsequently dried in a hot air oven (Universal oven Memmert UN 55, Schwabach, Germany) at 50 °C for 1 h. Then, it was continuously macerated in 95% ethanol at the same ratio mentioned above for 48 h (3 cycles). The filtrate solution was concentrated using a vacuum evaporator (Eyela, Tokyo, Japan) until dry to obtain *C. chinensis* seed ethyl acetate extract (DEA) and *C. chinensis* seed ethanol extract (DE), respectively. All extracts were kept in a tight container and stored at 4 °C until use. The percentage yield was obtained using the following equation:%yield = (E/P) × 100(1)
where E is the weight of the obtained extract and P is the weight of the plant powder used for extraction.

#### 2.2.2. Chemical Marker Analysis Using High-Performance Liquid Chromatography (HPLC)

Kaempferol and hyperoside were used as chemical markers in the *C. chinensis* extracts. The contents of the chemical markers were analyzed in both extracts using HPLC (Shimadzu Prominence, Tokyo, Japan). The HPLC condition was modified according to the method described by Liu et al. [21]. An HPLC analysis was performed on a C-18 column (KNAUER^®^ 250 × 4.6 mm, Berlin, Germany) as the stationary phase. The mobile phase consisted of acetonitrile and 0.1% *w/v* acetic acid with a gradient elution (0–13 min; 80%:20%, 13–18 min; 70%:30%, 18–26 min; 50%:50%, 26–30 min; 40%:60%, 30–31 min; 80%:20%, and 31–41 min; 80%:20%). The flow rate and injection volume were 1.0 mL/min and 10 µL, respectively. The HPLC chromatogram was detected using a UV detector at 360 nm. The experiment was performed in triplicates.

#### 2.2.3. Antioxidant Activity Determination

##### DPPH Radical Scavenging Assay

A DPPH assay was used to investigate the radical scavenging activity of the *C. chinensis* seed extracts following the method described by Phumat et al. [22]. Samples were prepared with various concentrations. Briefly, 20 µL of the sample was combined with 180 µL of 167 µM DPPH dissolved in ethanol. The sample was placed in a dark environment at room temperature for 30 min, and the absorbance was measured at 520 nm using a microplate reader (SpectraMax M3, San Jose, CA, USA). The DPPH solution without the sample was used as the control. The assay was performed in triplicate. Trolox was used as a positive control. The percentage of radical scavenging activity was determined using the following equation:% Inhibition = [(Ap − An) − (As − Ab)/(Ac − Ab)] × 100(2)
where Ap is the absorbance of the positive control containing the solvent mixed with DPPH, An is the absorbance of the negative control containing the solvent alone, As is the absorbance of the tested extract with DPPH, and Ab is the absorbance of the tested extract with the solvent.

The half maximal inhibitory concentration (IC_50_) was calculated from the plotted linear graph of sample concentration versus the percentage of inhibition.

##### Ferric Reducing Antioxidant Power (FRAP) Assay

A FRAP assay was used to evaluate the reducing properties of the extracts following the method described by Phumat et al. [22]. The FRAP reagent was a 10:1:1 mixture of 0.3 M acetate buffer (pH 3.6), 10 mM 2,4,6-Tripyridyl-S-triazine (TPTZ) dissolved in 40 mM of 37% *v/v* hydrochloric in deionized water, and 20 mM ferric chloride solution. Each extract was diluted in ethanol to a concentration of 1 mg/mL, and 20 µL of the sample was added to a 96-well plate and reacted with 180 µL of the FRAP reagent for 5 min at room temperature. The sample was measured at 595 nm using a microplate reader (SpectraMax M3, San Jose, CA, USA). The assay was performed in triplicate. Trolox was used as a positive control. An analytical curve was plotted using ferrous sulfate at different concentrations as a standard curve. The results are presented as mg of ferrous sulfate equivalent per gram of extract or FRAP values obtained from the following equation:FRAP value (mg FeSO_4_/g of extract) = (c × V × D)/N (3)
where c is the concentration of ferrous sulfate (mg), V is the sample volume (mL), D is the dilution factor, and N is the weight of the sample (g).

##### Lipid Peroxidation Inhibition Assay

The lipid peroxidation process was detected using the ferric-thiocyanate technique, as described by Phumat et al. [22]. The extracts were prepared at different concentrations in ethanol. Briefly, 0.3 mL of the sample was mixed with 1.40 mL of 1.3% *w/v* linoleic acid in methanol, 1.40 mL of pH 7.0 phosphate buffer (PBS), 0.70 mL of water, and 0.2 mL of 2,2′-azobis-(2-amidinopropane dihydrochloride) or AAPH in PBS. After incubation at 50 °C in a water bath (Memmert Waterbath WNB, Schwabach, Germany) for 4 h, 2.5 µL of the sample with linoleic acid and AAPH was mixed with 250 µL of 75% *v/v* methanol, 2.5 µL of ferrous chloride, and 2.5 µL of 10% *w/w* ammonium thiocyanate. After 5 min of incubation, the sample was measured at 500 nm using a microplate reader (SpectraMax M3, San Jose, CA, USA). The assay was performed in triplicate. Trolox was used as a positive control. The percentage of inhibition was calculated using Equation (2). The half maximal inhibitory concentration (IC_50_) was calculated from the plotted linear graph of sample concentrations versus the percentage of inhibition.

#### 2.2.4. Antibacterial Activity Determination

The antibacterial activities of the *C. chinensis* seed extracts were evaluated for the following three reference bacterial strains from the American Type Culture Collection (ATCC): *Enterococcus faecalis* (ATCC29212), *Pseudomonas aeruginosa* (ATCC 27853), and *Escherichia coli* (ATCC 25922). The antimicrobial assays were performed using agar disc diffusion and the minimum inhibitory concentration (MIC).

##### Agar Disc Diffusion Method

The antibacterial activities of the *C. chinensis* seed extracts were additionally assessed following the approved agar diffusion method suggested by the Clinical Laboratory Standard Institute (CLSI), 2020 [23]. Briefly, all strains were grown using nutrient agar (NA) for 18–24 h at 37 °C. The tested bacterial suspension was adjusted to obtain a turbidity of 0.5 McFarland nephelometer standard (1.5 × 108 CFU/mL). Each suspension was spread over Mueller Hinton agar (MHA) using sterile cotton swabs. For disc preparation, 20 µL of each extract was dropped onto a 6-mm-diameter sterile blank paper disc, then placed onto the MHA containing the tested bacteria and incubated at 37 °C for 24 h. Dimethyl sulfoxide (DMSO) was used to dissolve the extracts in a concentration range of 6.25–100 mg/mL. Tetracycline (30 µg/mL) and 100% of DMSO were used as the positive and negative controls, respectively. The inhibition zone was measured and was expressed as millimeters (mm). The assay was performed in triplicate.

##### Minimum Inhibitory Concentration (MIC) Determination

The *C. chinensis* seed extracts that exhibited antimicrobial activity using the agar disc diffusion method were selected for determining the MIC using the broth microdilution method [24]. The bacterial suspension was cultured in NA at 37 °C for 24 h. The bacterial inoculum was diluted to obtain a turbidity equivalent to the 0.5 McFarland nephelometer standard. Subsequently, 100 µL of cation-adjusted Mueller Hinton broth (CAMHB) was added to a 96-well plate. The extracts were then added at concentrations ranging from 25 to 0.098 mg/mL to the CAMHB. DMSO and tetracycline were used as a negative control and a positive control, respectively. Then, 100 µL of each bacterial suspension (*E. faecalis*, *P. aeruginosa*, and *E. coli*) was added to the 96-well plate and incubated at 37 °C for 18 h. The growth of the bacteria was observed after the incubation period. A clear solution indicated no bacterial growth, whereas a turbid solution indicated bacterial growth.

#### 2.2.5. Anti-Inflammatory Activity Determination

##### Cell Culture

The murine macrophage cell line (RAW 264.7) was maintained in Dulbecco’s modified Eagle medium (DMEM) supplemented with 10% *v/v* fetal bovine serum (FBS) and 1% *w/v* penicillin/streptomycin at 37 °C, 5% CO_2_, in a CO_2_ incubator (Thermo Fisher Scientific, Oxford, UK).

##### Cytotoxicity Test

An MTT assay was used to assess the cell viability of the *C. chinensis* seed extracts using RAW264.7 macrophage cells [25]. Briefly, the cells were seeded at 1 × 10^4^ cells/well in a 96-well plate in a CO_2_ incubator (Thermo Fisher Scientific, Oxford, UK) at 5% CO_2_. After 24 h of incubation, the cells were exposed to the sample at various concentrations and then cultured for a further 48 h. A total of 15 µL of the MTT solution at a concentration of 5 mg/mL was added and allowed to incubate for 4 h. Formazan crystals were dissolved with DMSO after removing the MTT solution and measured at 570 nm using a microplate reader (SpectraMax M3, San Jose, CA, USA). The percentage of cell viability was assessed using the following equation:% Cell viability = (OD of sample well/OD of vehicle control) × 100 (4)
where the OD of the sample is the absorbance of the treated cells, and the OD of the vehicle control is the absorbance of the untreated cells.

##### IL-1β, IL-6, and TNF-α Expression

A real-time polymerase chain reaction (RT-PCR) was used to assess the expression levels of the inflammatory cytokines, including IL-1β, IL-6, and TNF-α. Briefly, RAW 264.7 cells (10^6^ cells/mL) were seeded in a 12-well plate and cultured for 24 h. After pre-treatment with the extracts for 20 h, the cells were stimulated with lipopolysaccharide (LPS) for 4 h and harvested for the detection of RNA expression. The total RNA was extracted using an RNA extraction kit (PureLink RNA Mini Kit, Invitrogen, Waltham, MA, USA). The concentration of total RNA was estimated using a Nano-Drop spectrophotometer (Thermo Fisher Scientific, Oxford, MS, USA). Then, the RNA was converted to cDNA using the reverse transcriptase enzyme and an Omiscript RT Kit (QIAGEN, Hilden, Germany). The amount of β-actin, IL-1β, IL-6, and TNF-α was determined using RT-PCR. Briefly, 3 µL of cDNA was used with specific primers for β-actin, IL-1β, IL-6, and TNF-α, as shown in Table 1. The RT-PCR products were examined using electrophoresis on a 1.5% *w/w* agarose gel with 1X Tris acetate EDTA (TAE) buffer. The band intensities of the products were measured using a Gel Documentation and System Analysis machine (Transilluminator Boi View, Bio Step, Jahnsdorf, Germany). This method was used to represent the expression levels of IL-1β, IL-6, and TNF-α. Subsequently, the ratios relative to the expression levels of the control gene (β-actin) were calculated.

#### 2.2.6. In Vitro Irritation Test Using Hen’s Egg Chorioallantoic Membrane (HET-CAM) Assay

The effects of the *C. chinensis* seed extracts on acute irritation were examined using a HET-CAM assay [26]. The extracts were dissolved in water at a concentration of 5 mg/mL. The hen’s egg chorioallantoic membrane is a tissue that envelops the developing embryo of a chicken egg, containing a rich network of blood vessels. The chorioallantoic membrane was obtained from fertilized chicken eggs at 7–9 days of development. The procedure included cutting away the air sac portion using a rotating dentist saw blade and gradually peeling off the eggshell. Subsequently, the egg was treated with a normal saline solution and placed in an incubator for 15 min. The inner layer of the eggshell was carefully removed using forceps, and the test substances (30 µL) were then dropped onto the chorioallantoic membrane. The irritation reactions, namely, vascular hemorrhages, vascular lysis, and vascular coagulation, were investigated using a stereo microscope within 5 min (short term) and 60 min (long term). In this study, a normal saline solution was used as a negative control, and 1% *w/v* sodium lauryl sulfate was used as a positive control. The irritation score (IS) was calculated using the following equation:IS = (301 − *t*(*h*))/300 × 5) + (301 − *t*(*l*))/300 × 7) + (301 − *t*(*c*))/300 × 9)(5)
where *t*(*h*) is the time (s) at which the first vascular hemorrhages occurred, *t*(*l*) is the time (s) at which the first vascular lysis occurred, and *t*(*c*) is the time (s) at which the first vascular coagulation occurred. The results were classified as no irritation (IS = 0.0–0.9), slight irritation (IS = 1.0–4.9), moderate irritation (IS = 5.0–8.9), and severe irritation (IS = 9.0–21.0). Photographs of the blood vessels on CAM were taken under a stereo microscope.

#### 2.2.7. Molecular Docking

This study investigated the mechanisms of action of kaempferol and hyperoside, active biomarkers in *C. chinensis*, against inflammatory cytokines, such as IL-1β, IL-6, and TNF-α, using molecular docking. The 3D chemical structures of the bioactive markers were retrieved from the PubChem database and optimized at the molecular level using the HF/6–31 g(d,p) method. The crystal structures of IL-1β (PDB ID: 5fuc) [27], IL-6 (PDB ID: 8c3u) [28], and TNF-α (PDB ID: 2az5) [29] were obtained from the RCSB database. Docking simulations were performed using AutoDock Vina 1.2.0 [30]. The conformations with the lowest binding energies were further analyzed for the binding mode. The graphical visualization was generated using PyMOL 2 [31].

#### 2.2.8. Development of *C. chinensis* Seed-Extract-Loaded Nanoemulsions

##### Solubility Study

The solubility of the *C. chinensis* seed extracts (10 mg) was evaluated by dissolving them in various solvents, including oils (avocado oil, argan oil, and grape seed oil) and emulsifiers (Transcutol^®^ HP, Tween 80, polyglyceryl-3 polyricinoleate, and Cremophor^®^ RH40). Briefly, 10 mg of each extract was weighted in a test tube. Then, 10 µL of each solvent was added and mixed using a vortex mixer (Vortex-Genie^®^ 2; Scientific Industries, New York, NY, USA) for 1 min. Each solvent was continuously added until the extract completely dissolved. The solubility of each extract was indicated following British Pharmacopoeia 2022 as follows: very soluble means that 1 part of the sample can dissolve in less than 1 part of the solvent; freely soluble means that 1 part of the sample can dissolve in 1–10 parts of the solvent; soluble means that 1 part of the sample can dissolve in 10–30 parts of the solvent; sparingly soluble means that 1 part of the sample can dissolve in 30–100 parts of the solvent; slightly soluble means that 1 part of the sample can dissolve in 100–1000 parts of the solvent; very slightly soluble means that 1 part of the sample can dissolve in 1000–10,000 parts of the solvent; and insoluble means that 1 part of the sample can dissolve in greater than 10,000 parts of the solvent.

##### Preparation of *C. chinensis* Seed-Extract-Loaded Nanoemulsions

The compositions of the nanoemulsions are shown in Table 2. Avocado oil was used as the oil phase. Tween 80 and Transcutol^®^ HP were used as the surfactant and co-surfactant, respectively. The nanoemulsions were prepared using the high-energy method. The aqueous phase and oil phase were preheated at 75 °C and 70 °C on a hotplate stirrer, respectively. The aqueous phase was mixed with the oil phase using a high-shear homogenizer (IKA Werke GmbH & Co. KG, Staufen, Germany) at 6000 rpm for 5 min. The pre-emulsions were continually developed into nanoemulsions using a probe ultrasonic homogenizer (Vibra cell™, Sonics & Materials, Newtown, CT, USA) at 400 W and 40% amplitude, pulsed on and off every 3 s for 10 min. Then, all formulations were cooled down to room temperature for further use. The percentage and ratio of the surfactant to the co-surfactant were investigated for the appropriate nanoemulsions.

After obtaining the best nanoemulsions (small droplets size, polydispersity index (PDI) of less than 0.3, and good stability), 0.3% *w/w C. chinensis* seed extract was loaded into the selected nanoemulsions.

##### Characterization of *C. chinensis* Seed Extract-Loaded Nanoemulsions

The physical appearance of the formulations was investigated. Additionally, the particle size, polydispersity index (PDI), and zeta potential of the formulations were determined using photon correlation spectroscopy (Zetasizer ZS, Malvern Instruments Ltd., Malvern, UK) at 25 °C and a scattering angle of 173°. The samples were diluted with deionized water at a ratio of 1:100 before evaluation. The pH values of the formulations were measured at 25 °C using a calibrated pH meter (HICON, New Delhi, India).

##### Morphology of *C. chinensis* Seed Extract-Loaded Nanoemulsions Determined Using a Transmission Electron Microscope (TEM)

The morphology of the *C. chinensis* seed extract-loaded nanoemulsions was determined using a transmission electron microscope (TEM; Topcon, Tokyo, Japan). Briefly, 10 µL of the sample was dropped onto a 200-mesh copper grid (Sigma Aldrich, St. Louis, MO, USA) and left until dry. Then, phosphotungstic acid was dropped on top of the copper grid and allowed to dry for 30 min. The sample’s morphology was then viewed under the TEM.

##### Entrapment Efficiency of *C. chinensis* Seed Extract-Loaded Nanoemulsions

The entrapment efficiency of the *C. chinensis* seed extract-loaded nanoemulsions was evaluated using the ultrafiltration method [32]. First, 0.3 g of the formulation was dissolved in hexane and ethanol at a ratio of 6:4. The number of chemical markers in the sample was determined using HPLC [W_total_]. Conversely, the formulation (3 g) was added to the top of an ultra-centrifuge filter (Amicon^®^ Ultra 1.5 mL, NMWCO 10 kDa, Merck, Darmstadt, Germany). Then, the ultra-centrifuge filter was centrifuged at 7000 rpm and 25 °C for an hour. The unentrapped extract dropped to the bottom of the tube and was analyzed using HPLC [W_free_]. The percentage of entrapment efficiency (%EE) was calculated using the following equation:%EE = [W_total_] − [W_free_]/[W_total_] × 100(6)
where [W_total_] is the amount of total extract in the formulation and [W_free_] is the amount of unentrapped extract.

##### Stability Study of *C. chinensis* Seed Extract-Loaded Nanoemulsions

The *C. chinensis* seed extract-loaded nanoemulsions were kept in amber glass bottles under various storage conditions, including 30 °C with light protection for 30 days or heating and cooling (HC) for 6 cycles (the formulation was kept at 4 °C for 48 h and then moved to 45 °C for 48 h as 1 cycle). The physical appearance, including phase separation, creaming, or sedimentation, was observed. The particle size, PDI, zeta potential, and pH of the formulations were evaluated and compared before and after the stability study. All of the experiments were conducted in triplicate.

#### 2.2.9. Preparation of *C. chinensis* Seed Extract-Loaded Nanoemulsion-Based Gel

A *C. chinensis* seed extract-loaded nanoemulsion-based gel was prepared using a *C. chinensis*-seed-extract-loaded nanoemulsion to gel base ratio of 1:1. The gel base was composed of an ammonium acryloyldimethyltaurate/vinylpyrrolidone copolymer as a gelling agent. This formulation was used for in vivo wound care in animals.

#### 2.2.10. In Vivo Wound Care Using *C. chinensis* Seed Extract-Loaded Nanoemulsion-Based Gel

##### Experimental Animals

The in vivo wound care experiment was approved by the Ethics Committee of the Laboratory Animal Center, Chiang Mai University, Chiang Mai, Thailand (approval numbers 2566/RT-0018), following the ethical principles of the Association for Assessment and Accreditation of Laboratory Animal Care International (AAALAC). Healthy two-month-old male Wistar rats (*Rattus norvegicus*) weighing 200 g were selected for this study. They were housed in clean filter-top cages under the following conditions: a 12/12 h light/dark cycle, a temperature of 21 ± 1 °C, humidity of 50 ± 10%, and noise less than or equal to 85 dB. They had access to a standard diet and reverse-osmosis water at all times.

##### Wound Care Model

The wound care model in this study was adapted from the method of Subramanian and coworkers [33]. The rats were randomly divided into three groups (*n* = 21). In the control group, a 0.9% normal saline was used for wound treatment (*n* = 7); in the treatment group, the *C. chinensis* seed extract-loaded nanoemulsion-based gel was used for wound treatment (*n* = 7); and, in the positive control group, 2% Fucidin cream was used for wound treatment (*n* = 7). The rats were initially anesthetized with 5% isoflurane and continuously maintained under inhalation anesthesia with 3–5% isoflurane throughout the procedure. A medical skin pen was used to mark a square with a 20 × 20 mm area on the back of the animals. The dorsal hair of the anaesthetized rats was shaved off. Afterward, a sterile scalpel blade was used to make an incision along the marked line to create a wound. The wound was then cleaned with a 0.9% normal saline solution. The wound size was measured by placing a transparent sheet with marked grid lines over it and then the wound was photographed. The *C. chinensis* seed extract-loaded nanoemulsion-based gel and Fucidin cream (0.5 g) were applied to the wound area once a day for 21 days. A transparent sheet was placed on the wound area, and the wound was sketched using a permanent pen to measure its size. Wound healing was evaluated using the percentage of wound contraction with the following equation:%Wound contraction = [(W_0_ − W_n_)]/(W_0_) × 100(7)
where W_0_ is the wound area on day 0 and W_n_ is the wound area on days 6, 12, 18, and 21.

#### 2.2.11. Statistical Analysis

All measurements were performed in triplicate. The results are expressed as mean ± S.D. A statistical analysis was conducted using SPSS Software, Version 17.0. The differences between groups were analyzed using a one-way analysis of variance (ANOVA), followed by Tukey’s test. The differences were considered significant at a *p*-value ≤ 0.05.

## 3. Results and Discussion

### 3.1. C. chinensis Seed Extraction

The *C. chinensis* extracts obtained using various solvents had distinct characteristics. All of the *C. chinensis* seed extracts were a dark brown semi-solid mass. The percentage yields of the *C. chinensis* seed extract obtained from different solvent extraction methods showed that the ethanolic *C. chinensis* seed extract (DE) had the highest yield (3.20 ± 0.29% *w*/*w*), followed by the ethyl acetate *C. chinensis* seed extract (DEA) (2.50 ± 0.03% *w*/*w*). The DE had a higher yield than the DEA, because the semi-polarity of ethanol could extract a broad spectrum of polar compounds, whereas semi-polar compounds were extracted with ethyl acetate using fractional extraction [34].

### 3.2. Chemical Markers Analysis Using High-Performance Liquid Chromatography (HPLC)

To identify the primary individual components in the *C. chinensis* seed extracts obtained through various solvent extractions, qualitative and quantitative HPLC was used, and the most common components in all of the extracts were determined to be kaempferol and hyperoside. The amount of kaempferol and hyperoside in the DEA and DE was determined using HPLC by comparing the retention time with the standards. As shown in Figure 1, kaempferol and hyperoside could be detected in the DE, whereas only kaempferol could be detected in the DEA. The amount of kaempferol in the DEA and DE was 69.77 ± 0.25 mg/g extract and 21.16 ± 0.12 mg/g extract, respectively. Additionally, the amount of hyperoside in the DE was 3.23 ± 0.02 mg/g extract. The chemical compounds found in *C. chinensis* include flavonoids, polysaccharides, alkaloids, steroids, volatile oils, and lignans. The main flavonoids found in *C. chinensis* seeds are hyperoside, kaempferol, quercetin, rutin, and isorhametin [35,36]. The presence of significant amounts and types of bioactive compounds in *C. chinensis* seed extracts was determined based on the solvent selected for the extraction process. Kaempferol and hyperoside have been reported to have several pharmacological activities, such as antioxidant, anti-inflammation, antimicrobial, and anti-cancer activities [37,38,39]. Therefore, they could be used as markers for a further quantitative analysis of the extracts.

### 3.3. Antioxidant Activity of C. chinensis Seed Extracts

The antioxidant activities of the *C. chinensis* seed extracts obtained from various solvent extraction methods are shown in Table 3. In this study, the antioxidant activity of the *C. chinensis* extracts was determined using three different antioxidant tests related to different oxidative mechanisms. In general, the antioxidant assays can be divided into electron transfer (ET)-based assays and hydrogen atom transfer (HAT)-based assays. This study used DPPH and FRAP assays, which are ET-based assays, and a lipid peroxidation inhibition assay, which is a HAT-based assay. The extracts’ ability to scavenge free radicals was evaluated using the DPPH assay. This ability is associated with the electron transfer mechanism [40]. In addition, a FRAP assay was employed to demonstrate the extracts’ reducing activity, specifically, their capability to convert ferric ion (Fe^3+^) to ferrous ion (Fe^2+^) [41]. The ferric thiocyanate technique was employed to examine the inhibitory effects of the extracts on lipid peroxidation [42]. The results showed that the DE exhibited more antioxidant activity via its free radical scavenging ability, which was measured using the DPPH assay (*p* < 0.05), than the DEA. In contrast, the DEA exhibited higher reducing properties and lipid peroxidation inhibition activity than the DE, with a FRAP value of 0.37 ± 0.01 mg FeSO_4_/g extract and an IC_50_ of 4.73 ± 0.18, respectively. Previous research has reported that kaempferol can scavenge hydroxyl radical, superoxide anion, and peroxynitrite, and that it can stop hydroxyl radical production by chelating ferrous or cuprous ions. In addition, it can stimulate anti-oxidant enzymes, such as hemeoxygenase-1, superoxide dismutase, and catalase [43,44,45]. The hydroxyl groups in the structure of kaempferol may contribute to the antioxidant properties [46]. These results correspond to the higher kaempferol content in the DEA. Nevertheless, kaempferol and hyperoside were found in the DE. Previous research has reported that hyperoside exhibits a wide range of pharmacological effects, including antioxidant activity, anti-cancer activity, neuroprotective activity, antimicrobial activity, and anti-inflammatory activity [47,48]. It can decrease the intracellular reactive oxygen species (ROS) levels and lipid peroxidation in *Saccharomyces cerevisiae* [49]. In addition, another study found that it can prevent oxidative stress-induced liver injury, decrease superoxidase dismutase levels, and increase malondialdehyde levels [50]. Therefore, the antioxidant activity of the DE and DEA might be due to the flavonoids in the extracts.

### 3.4. Antibacterial Activity of C. chinensis Seed Extracts

The antibacterial activity of the extracts was measured qualitatively and quantitatively against selected bacteria using the disc diffusion method and a broth microdilution assay. In the disc diffusion assay (Table 4), the DEA showed high activity against *E. faecalis*, *E. coli*, and *P. aeruginosa*, whereas the DE showed no activity against any of the tested bacteria. The DEA was used for the determination of the minimal inhibitory concentration (MIC) against *E. faecalis*, *E. coli*, and *P. aeruginosa* using a broth microdilution assay (Table 5). The MIC results indicate that the DEA exhibited a growth-inhibitory effect against *E. faecalis* and *E. coli*, with a MIC value of 6.25 ± 0.0 mg/mL. Conversely, the MIC value against *P. aeruginosa* was 8.3 ± 3.6 mg/mL.

In this study, the DEA showed inhibitory activity against both Gram-positive bacteria (*E. faecalis*) and Gram-negative bacteria (*E. coli* and *P. aeruginosa*). The Gram-positive and Gram-negative bacteria in this study are generally known to be pathogenic bacteria that cause skin, soft-tissue, and wound infections. A previous study using different extraction methods, consistent with this study, found that the aqueous extract of *C. chinensis* showed a high ability to inhibit *Staphylococcus aureus* (*S. aureus*), *Kocuria rosea*, and *P. aeruginosa*, at a concentration of 100 mg/mL, using the agar disc diffusion method [51]. It is interesting that the antibacterial activity of ethanol and ethyl acetate *C. chinensis* seed extracts was first reported in this study. Most of the studies on antibacterial activity focus on other species [52,53]. In agreement with our study, a crude ethanolic extract of *Cuscuta reflexa* exhibited strong inhibition activity against similar pathogenic bacteria [54]. In contrast, the growth of *E. coli* was inhibited by *C. reflexa* at a concentration of 500 µg/mL [55]. A study conducted by Biswas et al., 2012, obtained better MIC values than the current study when examining the antibacterial activity of a methanolic *Cuscuta epithymum* extract against *P. aeruginosa* and *E. coli* (3.03 ± 0.16 and 3.47 ± 0.20) [56]. *Cuscuta* spp. have been reported to contain several chemical phytoconstituents that exhibit antimicrobial activity, such as flavonoids, alkaloids, tannins, phenolic compounds, and steroids [52]. Flavonoids have been reported to be important phenolic compounds for antibacterial activity [57]. Different antibacterial mechanisms of these compounds have been reported, such as cell membrane damage, the inhibition of nucleic acid synthesis, and the inhibition of the bacterial respiratory chain [58,59].

### 3.5. Cytotoxicity Test

An MTT assay was used to evaluate the effect of the *C. chinensis* seed extracts on RAW264.7 cell viability, as shown in Figure 2. The cells were exposed to various concentrations of the extracts for 24 h. The findings indicate a cell viability of more than 60% with the DEA and DE in a concentration range of 63–125 µg/mL. Conversely, a cell viability of more than 50% was found with the DEA and DE in a concentration range of 250–1000 µg/mL. Thus, a non-toxic concentration of the DEA and DE at 125 μg/mL was chosen for the anti-inflammatory activity study.

### 3.6. The Effects of C. chinensis Seed Extracts on Inhibiting Gene Expression of Inflammatory Cytokines

The effects of the *C. chinensis* extracts on the mRNA expression levels of IL-1β, IL-6, and TNF-α were determined using RT-PCR. As shown in Figure 3, Figure 4 and Figure 5, the treatment with LPS significantly increased the transcription of all cytokine genes and was used as a positive control. The results showed that pre-treating the cells with the DE and DEA extracts at a concentration of 125 µg/mL was able to suppress the LPS-induced expression of IL-1β, IL-6, and TNF-α gene levels when compared to the positive control (*p* < 0.05). Interestingly, the DE decreased the expression levels of IL-1β, IL-6, and TNF-α better than the DEA, with a significant difference (*p* < 0.05). Notably, these results are consistent with those of a study conducted by Kang et al., 2014, which reported that *C. chinensis* extracts had the ability to suppress the production of TNF-α, IL-1β, and IL-6 by down regulating their transcription levels [12]. Kim et al., 2019, also reported that *C. chinensis* extracts reduced the transcriptional levels of inflammatory factors, including TNF-α, IL-6, and IL-1β, in mice [13]. Another study investigated the anti-inflammatory properties of a *C. chinensis* extract in mice when administered at concentrations of 100 and 500 mg/kg. It was found that the extract helped to reduce swelling in the mice’s feet. This effect may be attributed to the extract’s ability to decrease nitric oxide and malondialdehyde levels. Furthermore, the extract also aided in reducing the secretion of IL-1β, IL-6, NF-kB, TNF-α, and COX-2 [11]. Therefore, our study suggests that the DE and DEA can suppress inflammation by reducing the expression of genes associated with the inflammatory response.

Inflammation is one of the stages of wound healing. It is a key stage and plays a vital role in protecting the injured tissues from pathogens and in stimulating the progression of wound healing. The appropriate modulation of inflammation during healing is important in preventing chronic wound development and achieving complete wound healing [60]. The inflammatory microenvironment of skin wounds consists of inflammatory cells (neutrophils, macrophages, mast cells, Langerhans cells, T lymphocytes, platelets, and complement) and non-inflammatory cells (keratinocytes, fibroblasts, and vascular endothelial cells) [61]. IL-1β, IL-6, and TNF-α play a crucial role in wound healing. IL-1β and TNF-α can be found in the homeostasis, inflammatory, and proliferation phases, whereas IL-6 is related to the early inflammatory phase [62]. Previous research has reported the anti-inflammatory activity of flavonoids. The hydroxyl groups in their structure can exert antibacterial, antioxidant, and anti-inflammatory effects due to high hydroxylation levels [63]. Kaempferol has the potential to increase the wound healing rate and increase the expression of collagen type 3, vascular endothelial growth factor, and basic fibroblast growth factor in wound granulation tissue [33,64,65]. Hyperoside has also demonstrated anti-inflammatory activity by inhibiting inflammatory pathways and repairing DNA damage [66]. Additionally, it can reduce the IL-1β and TNF-α contents in rat cerebral ischemia–reperfusion injury models [67].

### 3.7. Molecular Docking

In order to understand the mechanism underlying the anti-inflammatory activity, binding studies of the major markers in *C. chinensis* were performed through molecular docking. The docking results showed that hyperoside and kaempferol exhibited preferable binding to IL-1 β, IL-6, and TNF, with binding scores of less than −6.0 kcal/mol.

Regarding the binding to IL-1, kaempferol exhibited the highest docking score at −7.3 kcal/mol, indicating a superior affinity to hyperoside (−7.0 kcal/mol). Hommel et al. elucidated the binding pocket of IL-1β, situated between the β4–5 loop and the β7–8 loop [27], which is the site where kaempferol and hyperoside demonstrated a good fit in our docking analysis. The binding interactions of these two compounds and the receptor involve hydrogen bond formations with residues of Ser45 and Lys94 (Figure 6a). In contrast, hyperoside exhibited a potent binding affinity (−7.3 kcal/mol) toward IL-6, and it was greater than that of kaempferol (−6.7 kcal/mol).

The mechanism of IL-6 involves the binding of the IL-6 ligand to its soluble receptor (IL-6R or gp180), followed by binding to gp130. Our docking simulations suggest that both hyperoside and kaempferol are present in the cavity at the interface of the binding of IL-6 [28]. Hyperoside can form six hydrogen bonds with the residues of Arg104, Pro107, Pro162, Gly164, and Gln190, while kaempferol forms four hydrogen bonds with Arg104, Pro107, Gly164, and Ala192 (Figure 6b). The strong binding affinity of hyperoside might be related to the hydrogen formation of the galactoside moiety against IL-6R. Similar to its effect on the IL-6 receptor, hyperoside exhibited a stronger binding affinity toward the TNF-α receptor than kaempferol, with binding affinities of −7.5 and −7.3 kcal/mol, respectively. Hyperoside forms three hydrogen bonds with the residues of Ser60, Gly121, and Tyr151, while kaempferol also forms hydrogen bonds with Leu120 and Tyr151 (Figure 6c).

Molly et al. investigated the small-molecule inhibitors bound to the interface be-tween domain A and domain B of the TNF-α receptor [29]. Our docking simulations demonstrate that hyperoside and kaempferol bind within the same region of the TNF-α inhibitor. Previous studies support our molecular docking findings. Hyperoside has been shown to significantly reduce the levels of IL-1β, IL-6, and TNF in type 2 diabetic rats [68]. Additionally, hyperoside has been found to indirectly suppress IL-1β and TNF-α production in LPS-stimulated microglial cells [69] and mitigate IL-1β-induced ROS production and apoptosis in chondrocytes [70]. Kaempferol has been shown to suppress the levels of IL-1β, IL-6, and TNF in LPS-induced inflammation in cardiac fibroblasts [71]. In addition, kaempferol has been found to inhibit IL-1β-induced osteoclastogenesis and attenuate IL-1β-induced inflammatory responses in skin fibroblasts [72].

### 3.8. Irritation Assessment Using Hen’s Egg Test on the Chorioallantoic Membrane (HET-CAM) Assay

In the last decade, the HET-CAM test has been extensively utilized to evaluate the irritation of substances. As shown in Table 6, the positive control, 1% *w/v* SLS, caused severe irritation, with an irritation score (IS) of 11.9 ± 0.6; whereas the negative control, 0.9% *w/v* NaCl, caused no irritation. After 60 min of exposure to the positive control, all signs of irritation, including bleeding, coagulation, and vascular lysis, were observed, as shown in Figure 7. Interestingly, no signs of irritation were observed on the CAM after exposure to the DE and DEA extracts, even after 60 min. Although ethanol and ethyl acetate may lead to skin irritation and contact dermatitis, all of the solvents were completely removed from the *C. chinensis* seed extracts through the process of evaporation.

### 3.9. Development and Characterization of Unloaded Nanoemulsions and C. chinensis Extract-Loaded Nanoemulsions

The DE was chosen for loading into nanoemulsions, due to its high percentage yield with a green extraction solvent and its good antioxidant and anti-inflammatory activities. Moreover, the DE contained effective flavonoids, such as kaempferol and hyperoside, which indicated a good ability for wound healing. The good solubility of the extract in the formulation is an important factor for maintaining the extract in a solubilized form in nanoemulsions. The DE is more soluble in avocado oil (10 mg/0.2 mL) than in argan oil (10 mg/0.30 mL) and in grape seed oil (10 mg/mL). In addition, the DE is soluble in different surfactants, as follows: Transcutol^®^ HP (10 mg/0.10 mL) equal to Tween 80 (10 mg/0.10 mL), followed by polyglyceryl-3 polyricinoleate (10 mg/0.50 mL) and Cremophor^®^ RH40 (10 mg/mL). The nanoemulsions are composed of oil, surfactant, co-surfactant, and water. Based on these results, avocado oil was selected as the oil phase for nanoemulsion preparation. Tween 80 and Transcutol^®^ HP were selected as the surfactant and co-surfactant, respectively, with the aim of formulating o/w nanoemulsions. Tween 80 and Transcutol^®^ HP are non-ionic surfactants, with HLB values of 15 and 4.3, respectively. Moreover, in surfactant selection, the hydrophilic–lipophilic balance (HLB) value of the components must be considered to form o/w nanoemulsions. In a blended surfactant, an HLB value of greater than 10 is the minimum requirement of the criterion for stable o/w nanoemulsions [73]. A suitable blend of low and high HLB surfactants leads to the formation of stable nanoemulsions. The development of nanoemulsions depends on the percentage and ratio of S_mix_ (surfactant and co-surfactant).

In the present study, the optimization of nanoemulsions was achieved at a particle size below 180 nm. The achieved particle size was confirmed to be in the nano-range [74]. The S_mix_ ratios of 1:1, 2:1, and 3:1, with increasing Tween 80, likely caused reductions in particle size, as shown in Table 7. The unloaded NE F5 presented the smallest particle size at 88.42 ± 1.10 nm, with a narrow PDI value of 0.27 ± 0.01 and a good zeta potential of −20.90 ± 0.56 mV. As it presented a narrow PDI (less than 0.3) with a small particle size, it could generate stable nanoemulsions. In addition, 0.3% *w/w* DE was loaded into the unloaded NE F5. The DE-loaded nanoemulsions, or DE-loaded NE, demonstrated a particle size of 136.80 ± 1.95 nm, a PDI value of 0.25 ± 0.01, and a zeta potential of −20.90 ± 0.56 mV. The stability of the unloaded NE F5 and the DE-loaded NE was evaluated under various conditions.

The preparation methods of nanoemulsions, such as low-energy and high-energy methods, affect the particle size and the polydispersity index (PDI) [75]. Low-energy methods, such as the phase inversion temperature (PIT) and the phase inversion composition (PIC) methods, require internal chemical energy of the systems and gentle stirring for the production of the nanoemulsions [76]. Conversely, high-energy methods, such as high-pressure homogenization and ultrasonication, require strong forces with high kinetic energy supplied by mechanical equipment for the preparation of the nanoemulsions [77]. The nanoemulsions in this study were prepared using the ultrasonication method, with high energy provided by ultrasonic waves, which break down large-sized particles to nano-sized particles. Previous research has reported that high-energy methods affect the particle size and the PDI of nanoemulsions, as well as providing them with greater stability [78]. A nanoparticle size of between 70 and 180 nm, after preparation via ultrasonication, was reported in our results and those of Smruthi et al., 2022, and Nirmal et al., 2023 [79,80]. The PDI value indicates the size distribution, homogeneity, and stability of the nanoemulsions. The PDI, which ranges from 0 to 1, reflects the uniformity of the droplets in the formulation. A PDI lower than 0.2 is considered desirable, as it indicates a good particle size distribution [75,80]. The zeta potential, which describes the surface charge on the droplets, is crucial for stability, generating repulsion forces between the droplets to prevent particle agglomeration. The desirable zeta potentials are −30 mV to +30 mV, as they provide good stability [81].

#### 3.9.1. Morphology of Nanoemulsions and *C. chinensis* Extract-Loaded Nanoemulsions Determined Using Transmission Electron Microscope (TEM) Images

The morphology of the nanoemulsions was evaluated using TEM images, as shown in Figure 8. The obtained particles of the unloaded NE F5 and the DE-loaded NE were spherical in shape, had a smooth surface, were homogeneously colored, and showed no aggregation. In addition, they had a size of approximately less than 200 nm.

#### 3.9.2. Stability Study of Unloaded Nanoemulsions and *C. chinensis* Extract-Loaded Nanoemulsions

The stability of the unloaded NE F5 and the DE-loaded NE was then evaluated under various conditions, including heating–cooling (HC) for six cycles and 30 °C for 1 month. The results are shown in Table 8. The particle size of the unloaded NE F5 was mostly stable at 30 °C and slightly increased after the stability test. However, the particle size and the PDI value of the DE-loaded NE under HC conditions significantly increased compared to the initial values (*p* < 0.05). Conversely, the particle size and the PDI value did not change after storage at 30 °C for 1 month. The zeta potential of the DE-loaded NE increased after the stability study. However, the physical appearance and the pH value of the formulation did not change after the stability study. The formulation had a pH value of 5.5, which is appropriate for topical application. Thus, the DE-loaded NE was slightly affected by a high temperature, but no phase separation occurred.

#### 3.9.3. Entrapment Efficacy of *C. chinensis* Extract-Loaded Nanoemulsions

The entrapment efficiency represents the percentage of compounds entrapped in nanoparticles. The entrapment efficacy of kaempferol in the DE-loaded NE was found to be 96.71 ± 0.91%. The entrapment efficiency ability of the extracts is influenced by many factors, such as the amount of lipid and water, the solubility of compounds in lipid and water, and the concentration of the surfactant. The entrapment of extracts in the lipid or water phases depends on the solubility of each extract. Previous research has reported that increasing the concentration of the surfactant increases the entrapment efficiency of nanoemulsions [82]. The ratio of S_mix_ (3:1) can increase %EE by more than 90, due to the surfactant forming micelles capable of entrapping the bioactive compound inside of the nanoemulsion. In addition, the solubility of DE in the oil phase is an important factor in increasing the entrapment efficiency percentage of the DE in nanoemulsions.

### 3.10. Wound Care Effect of C. chinensis Extract-Loaded Nanoemulsion-Based Gel

The DE-loaded NE was prepared in a gel base to appropriately increase the viscosity for use on the wounds. The rats were divided into three groups, as follows: a negative control group (0.9% NaCl), a treatment group (DE-NE-based gel), and a positive control group (2% Fucidin cream). Photographs of the wound healing process in each group are shown in Figure 9. The results show that the treatment group exhibited an essential impact on the wound healing process relative to the untreated group (negative control). The wound contraction results are shown in Figure 10. The DE-NE-based gel increased the percentage of wound contraction more than the Fucidin cream at all time intervals (*p* < 0.05). The skin’s ability to heal itself depends on many factors, including the presence of infection, the dryness of the wound, and the individual skin healing function. Previous research has shown that normal saline can have a better effect on wound healing than tap water and antiseptic solution [83]. Normal saline can help to clean a wound and keep its surface moist during the healing process. In contrast, Fucidin cream contains fusidic acid, which is used to treat skin infections caused by bacteria. However, it did not accelerate the wound contraction process. The wound healing process consists of the following four separate stages: hemostasis, inflammation, proliferation, and remodeling. From the results outlined above, it was determined that the DE has the ability to reduce inflammation by decreasing the expression of pro-inflammatory markers, including IL6, IL-1β, and TNF-α, in macrophages. The decrease in IL-1β and TNF-α reveals a pro-healing phenotype and improves wound healing [84]. The DE has potential therapeutic targets to treat wounds during the inflammatory stage. In addition, the DE was evaluated in terms of the role of its constituents, kaempferol and hyperoside, which are flavonoids responsible for wound healing due to their anti-inflammatory effects. The effects of flavonoids on wound healing have been reported to occur through many pathways, such as growth factor-beta (TGF-β), nuclear factor kappa B (NF-κB), and nitric oxide (NO) [85]. Previous studies have reported that kaempferol and hyperoside accelerate wound healing and dermal remodeling [86]. These characteristics of flavonoids have a meaningful influence on wound care. Another study revealed that a living microalgal-based procedural hydrogel dressing possessed a synergistic effect of antibacterial, oxygen release, ROS scavenging, anti-inflammatory, and macrophage polarization regulation to promote rapid wound healing [87]. In addition, the development of compound-loaded hydrogel will help to enhance wound healing, antimicrobial properties, and biocompatibility [87]. Therefore, the DE-NE-based gel presented a positive effect for wound care.

## 4. Conclusions

Ethanol *C. chinensis* seed extract showed a higher percentage yield from maceration than ethyl acetate *C. chinensis* seed extract. It possessed radical scavenging activity and anti-inflammatory activity by reducing the expression of pro-inflammatory cytokines more than the ethyl acetate *C. chinensis* seed extract. In contrast, the ethyl acetate *C. chinensis* seed extract presented a better metal-reducing property, lipid peroxidation inhibition, and antibacterial activity than the ethanol *C. chinensis* seed extract. The flavonoids found in the extracts, especially kaempferol and hyperoside, were related to the biological activities of the extracts. *C. chinensis* seed extract-loaded nanoemulsions were successfully developed with a small particle size, a narrow PDI value, and good stability. Moreover, the *C. chinensis* seed extract-loaded nanoemulsion-based gel has good potential for wound care in rats. Therefore, this formulation can be further used as an alternative product for wound healing in clinical studies.

## Figures and Tables

**Figure 1 pharmaceutics-16-00573-f001:**
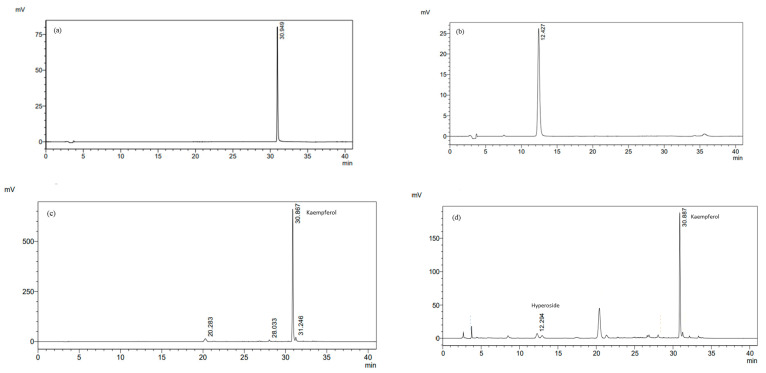
HPLC chromatograms of (**a**) kaempferol and (**b**) hyperoside at a final concentration of 10 µg/mL, and (**c**) DEA and (**d**) DE at a final concentration of 1 mg/mL detected at 360 nm.

**Figure 2 pharmaceutics-16-00573-f002:**
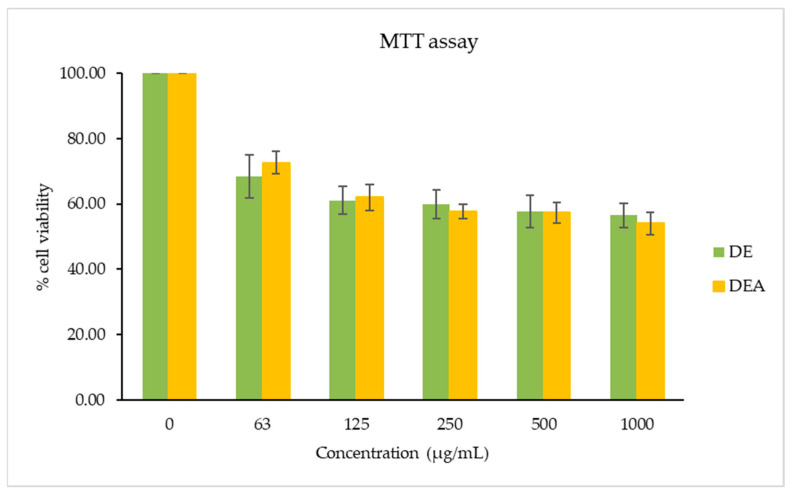
The cell viability of RAW264.7 cells treated with various concentrations of *C. chinensis* seed ethyl acetate (DEA) and *C. chinensis* seed ethanol (DE) extracts using MTT assays. The different letters above the columns of each extract indicate statistically significant differences (*p* < 0.05).

**Figure 3 pharmaceutics-16-00573-f003:**
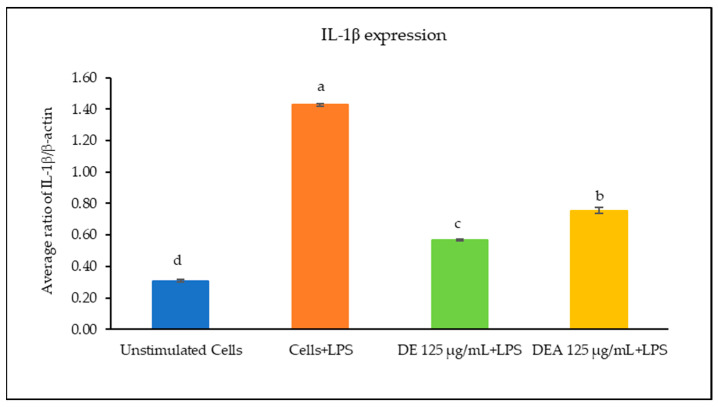
Ratio of IL-1β/β-actin of RAW 264.7 cells when treated with DE and DEA. The different letters above the columns of each extract indicate statistically significant differences (*p* < 0.05).

**Figure 4 pharmaceutics-16-00573-f004:**
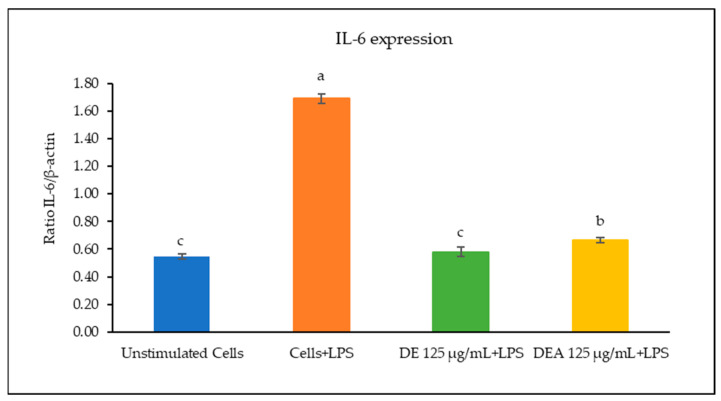
Ratio of IL-6/β-actin of RAW 264.7 cells when treated with DE and DEA. The different letters above the columns of each extract indicate statistically significant differences (*p* < 0.05).

**Figure 5 pharmaceutics-16-00573-f005:**
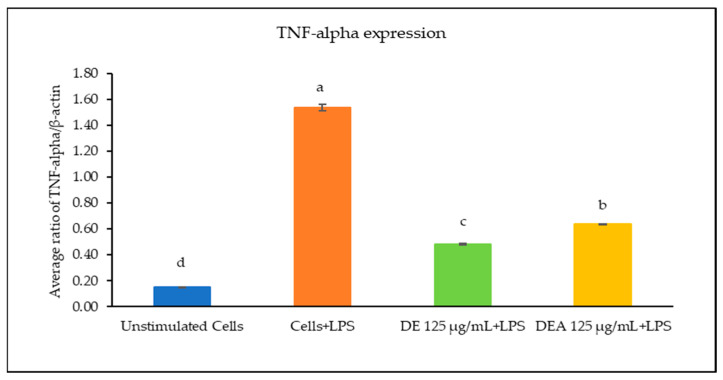
Ratio of TNF-α/β-actin of RAW 264.7 cells when treated with the DE and DEA. The different letters above the columns of each extract indicate statistically significant differences (*p* < 0.05).

**Figure 6 pharmaceutics-16-00573-f006:**
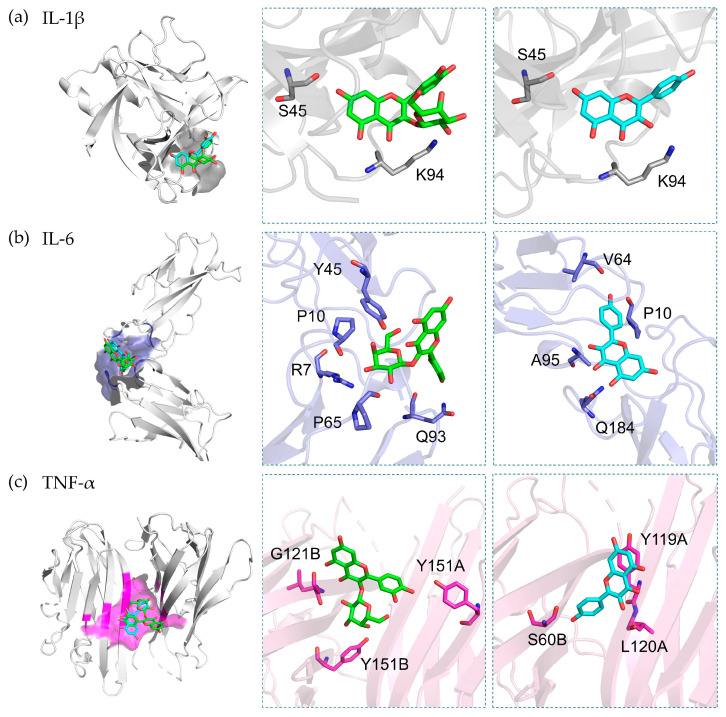
Binding modes of hyperoside (green) and kaempferol (blue) within the binding sites of (**a**) IL-1β (gray), (**b**) IL-6 (slate), and (**c**) TNF-α (magenta). The ligand and amino acid involved in binding are shown in stick style, and the proteins are shown in ribbon style.

**Figure 7 pharmaceutics-16-00573-f007:**
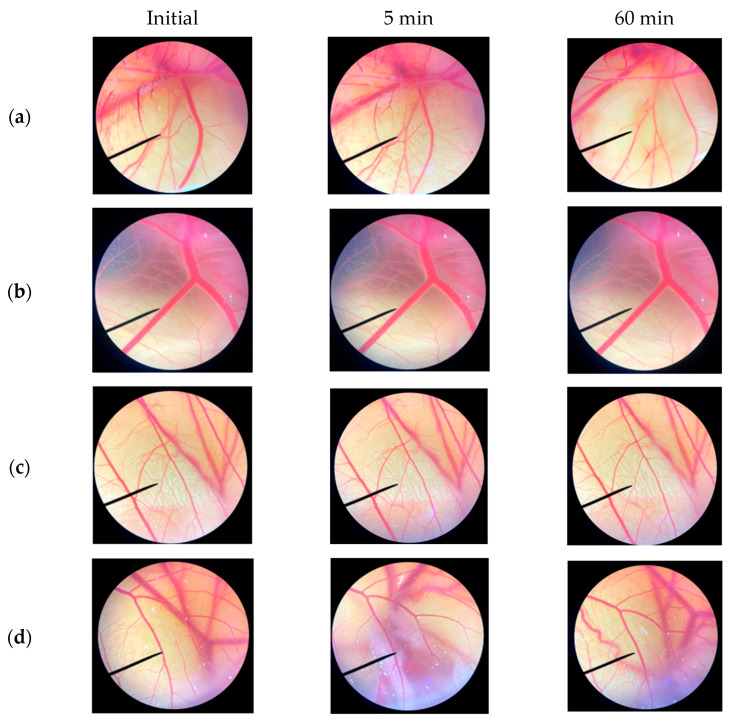
CAM initially and after exposure to (**a**) positive control (1% *w/v* SLS), (**b**) negative control (0.9% *w/v* NaCl), (**c**) DE (5 mg/mL), and (**d**) DEA (5 mg/mL) for 5 min and 60 min.

**Figure 8 pharmaceutics-16-00573-f008:**
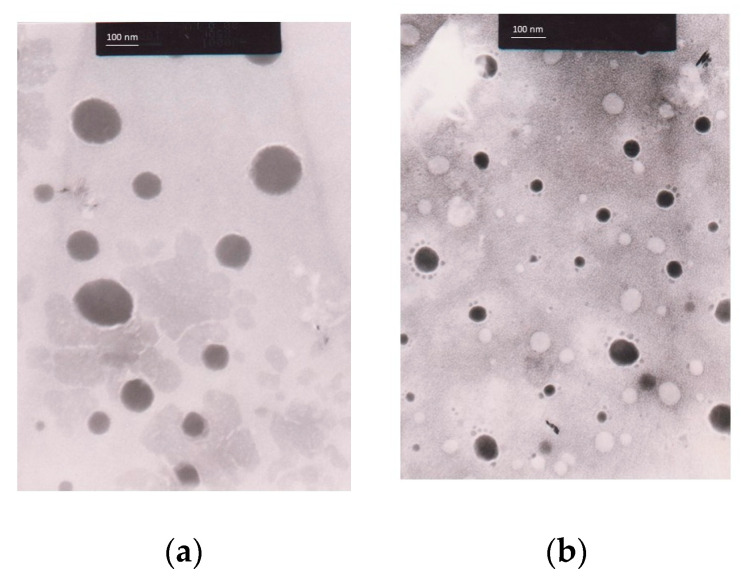
TEM images of (**a**) unloaded NE F5 and (**b**) DE-loaded NEs.

**Figure 9 pharmaceutics-16-00573-f009:**
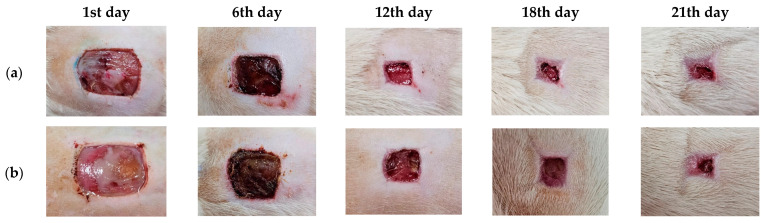
Photographs of wound area of (**a**) negative control group (0.9% NaCl), (**b**) treatment group (DE-NE-based gel), and (**c**) positive control group (2% Fucidin cream) on day 1 and on days 6, 12, 18, and 21.

**Figure 10 pharmaceutics-16-00573-f010:**
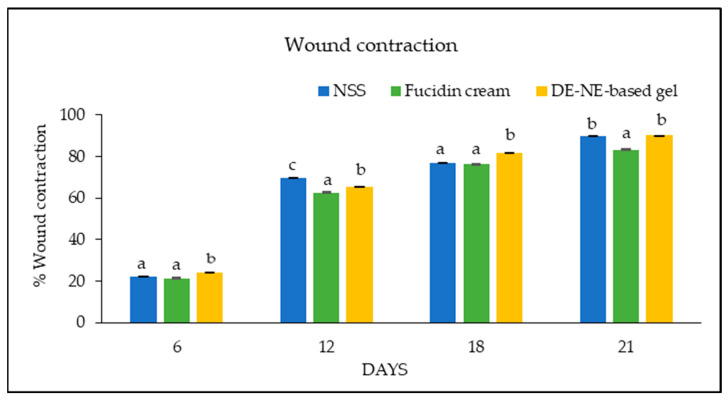
Percentage of wound contraction after treatment with 0.9% NaCl (NSS), DE-NE-based gel, and 2% Fucidin cream. The different letters above the bars indicate significant differences at *p* < 0.05.

**Table 1 pharmaceutics-16-00573-t001:** Oligonucleotide primers used for RT-PCR.

Gene	Primer	Sequence: (5′-3′)
β–actin	Forward Reverse	TCATGAAGTGTGACGTTGACATCCGT CCTAGAAGCATTTGCGGTGCACGATG
IL-1β	Forward Reverse	CAGGATGAGGACATGAGCACC CTCTGCAGACTCAAACTCCAC
IL-6	Forward Reverse	CATCCAGTTGCCTTCTTGGGA GCATTGGAAATTGGGGTAGGAAG
TNF-α	Forward Reverse	ATGAGCACAGAAAGCATGATC TACAGGCTTGTCACTCGAATT

**Table 2 pharmaceutics-16-00573-t002:** Compositions of unloaded nanoemulsions (Nes) (values are given as % *w/w*).

Compositions	NE1	NE2	NE3	NE4	NE5	NE6
Avocado oil	10	10	10	10	10	10
S_mix_ ratio	1:1	1:1	2:1	2:1	3:1	3:1
S_mix_ (%)	10	15	10	15	10	15
Water	80	75	80	75	80	75

S_mix_ is the surfactant and co-surfactant (Tween 80 and Transcutol^®^ HP).

**Table 3 pharmaceutics-16-00573-t003:** Antioxidant activity of *C. chinensis* seed extracts measured using DPPH, FRAP, and lipid peroxidation inhibition assays.

Samples	DPPH IC_50_ (µg/mL)	FRAP Value (mg FeSO_4_/g Extract)	Lipid Peroxidation Inhibition IC_50_ (µg/mL)
DEA	4.85 ± 0.18 ^a^	0.37 ± 0.01 ^b^	4.73 ± 0.18 ^b^
DE	1.57 ± 0.01 ^b^	0.20 ± 0.02 ^c^	6.29 ± 0.54 ^a^
Trolox	1.05 ± 0.06 ^c^	1.44 ± 0.04 ^a^	0.21 ± 0.01 ^c^

Different letters indicate significance level at *p* < 0.05.

**Table 4 pharmaceutics-16-00573-t004:** Antibacterial activity of the *C. chinensis* seed extracts determined using the disc diffusion method.

Concentration (mg/mL)	Inhibition Zone (mm.)									
	100		50		25		12.5		Tetracycline (30 µg/mL)	DMSO
Extracts/ Microbial	DE	DEA	DE	DEA	DE	DEA	DE	DEA		
*E. faecalis*	-	11 ± 1.0	-	10 ± 0.6	-	-	-	-	15 ± 1.7	-
*E. coli*	-	7 ± 0	-	-	-	-	-	-	30 ± 0	-
*P. aeruginosa*	-	9 ± 4.0	-	8 ± 3.2	-	-	-	-	24.3 ± 0.6	-

The disc diameter is 6 mm. “-” indicates no inhibition.

**Table 5 pharmaceutics-16-00573-t005:** Minimum inhibitory concentration (MIC) of the *C. chinensis* seed extracts determined using the broth microdilution assay.

Extracts /Microbial		MIC (mg/mL)	
	*E. faecalis*	*E. coli*	*P. aeruginosa*
DEA	6.25 ± 0.0	6.25 ± 0.0	8.3 ± 3.6

**Table 6 pharmaceutics-16-00573-t006:** Irritation score (IS) and irritation assessment of *C. chinensis* seed extracts using HET-CAM assay.

Samples	IS	Irritation Assessment
Positive control (1% *w/v* SLS)	11.9 ± 0.6	Severe
Negative control (0.9% *w/v* NaCl)	0.0 ± 0.0	No irritation
DE (5 mg/mL)	0.0 ± 0.0	No irritation
DEA (5 mg/mL)	0.0 ± 0.0	No irritation

**Table 7 pharmaceutics-16-00573-t007:** Particle size, PDI, and zeta potential of unloaded nanoemulsions.

Formulation	Particle Size (nm)	PDI	Zeta Potential (mV)
Unloaded NE F1	174.90 ± 2.34 ^d^	0.28 ± 0.04 ^c^	−18.03 ± 0.40 ^b^
Unloaded NE F2	211.27 ± 2.04 ^e^	0.37 ± 0.01 ^c^	−19.70 ± 0.44 ^b^
Unloaded NE F3	152.90 ± 1.21 ^c^	0.20 ± 0.03 ^a^	−0.25 ± 0.49 ^a^
Unloaded NE F4	124.13 ± 1.15 ^b^	0.25 ± 0.02 ^b^	−20.20 ± 0.26 ^b^
Unloaded NE F5	88.42 ± 1.10 ^a^	0.27 ± 0.01 ^b^	−20.90 ± 0.56 ^b^
Unloaded NE F6	119.37 ± 0.45 ^a,b^	0.26 ± 0.01 ^b^	−19.20 ± 0.78 ^b^

The different letters in each column indicate significant differences at *p* < 0.05.

**Table 8 pharmaceutics-16-00573-t008:** Particle size, PDI, and zeta potential of unloaded NE F5 and DE-loaded NE before and after the stability study.

	Conditions	Particle Size (nm)	PDI	Zeta Potential (mV)
Unloaded NE F5	Initial	132.04 ± 1.41 ^a^	0.27 ± 0.02 ^a^	−20.90 ± 0.56 ^a^
	HC	142.60 ± 1.20 ^b^	0.28 ± 0.00 ^a^	−26.60 ± 0.82 ^b^
	30 °C	134.27 ± 1.67 ^a^	0.26 ± 0.00 ^a^	−23.33 ± 0.85 ^a,b^
DE-loaded NE	Initial	136.80 ± 1.95 ^a^	0.25 ± 0.01 ^a^	−24.07 ± 0.25 ^b^
	HC	206.83 ± 2.62 ^b^	0.27 ± 0.01 ^b^	−21.20 ± 0.82 ^a^
	30 °C	131.07 ± 1.02 ^a^	0.26 ± 0.01 ^a^	−23.33 ± 0.85 ^a,b^

The different letters in each column indicate significant differences at *p* < 0.05.

## Data Availability

Data are contained within this article.

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
