# Peer review of "Exploring the Wound Healing Potential of a Cuscuta chinensis Extract-Loaded Nanoemulsion-Based Gel"

_pharmaceutics, 2024, doi:10.3390/pharmaceutics16050573_

Round 1
Reviewer 1 Report
Comments and Suggestions for Authors
The Cuscuta chinensis has several pharmacological functions, such as anti-oxidant, anti-inflammatory, antitumor, and anti-diabetic activity. The authors Nichcha et al. did several studies such as the extraction of phytochemical markers of Cuscuta chinensis, preparation of nanoemulsion-based gel for wound healing, etc. are appreciable. However, several minor and major corrections are required to improve the quality of this manuscript.
Minor:
1. In the introduction section, Lines 69 to 72, Write the full name of MDA in line number 69 instead of line number 72.
2. In the line 23, C. chinensis should also be written in the line 19. E.g. Cuscuta chinensis (C. chinensis) presents many pharmacological activities, including antidiabetic effects, 19 antioxidant, anti-inflammatory, and antitumor properties.)
3. In the line 101, Either write Cuscuta chinensis or C. chinensis.
4. In lines 133 and 138, the incorrect degree symbol is written. Correct it.
5. Line numbers 144 to 145, the full form of HPLC is already mentioned earlier, so there is no need to write the full form again; only write HPLC.
6. Line 151, in the sentence “The UV detector was detected at 360 nm.” Rewrite the sentence to improve the clarity.
7. Line 163, equation 2, write the minus sign in the middle.
8. Line 170, rephrase the sentence to improve the clarity.
9. Line number 188-191, write about the “mixed solution” the author is talking about. Rewrite the methodology to improve clarity.
10. In section 2.2.4, line 198, the Author mentioned the “four reference bacterial strains” but in the upcoming lines, only three are written. Correct it.
11. Line numbers 210 to 211, write the “concentration range” instead of “range concentration.”
12. Page number 5, Section 2.2.4, in the second paragraph (minimum inhibitory concentration determination)- The whole paragraph has grammatical inconsistencies and a few sentences are unclear/can be presented in an understandable form. For instance, Lines 224 to 226.
13. In the whole manuscript, give the space (if needed) between the digit and their unit of temperature, weight, and percentage. (e.g. 37°C change to 37 °C and 10%v/v change to 10% v/v).
14. Line number 240, what digit 570 represents, and write its unit too.
15. Section 2.2.5, under the IL-1β, IL-6, and TNF- Expression paragraph- The whole paragraph has grammatical inconsistencies, and a few sentences are unclear/can be presented in an understandable form. For instance, Lines 256 to 257. In the same heading, give a comma after IL-6, and in line number 263, give the space after the word “Germany.”
16. Line number 269, rewrite the sentence with grammatical improvement.
Major:
The whole manuscript has several grammatical errors, and various sentences are not clear. Some of them are commented on in minor, and some are in major but still need several corrections.
1. Page number 7, Section 2.2.8, lines number 300 and 306, rewrite the sentences with grammatical improvement and also write the solubility determination procedure used in this manuscript.
2. In section 3.4, line numbers 483 to 485, the authors reported that the MIC value of the antibacterial activity of methanolic Cuscuta epithymum extract against P. aeruginosa and E. 484 coli (3.03 ± 0.16 and 3.47 ± 0.20) better than the current study. Then why is the author using C. chinensis?
3. In section 3.9, first paragraph, how particle size was measured? Write the method and instrument name. In the second paragraph, what are the ultrasonic parameters like pulse on/off time, total induction time, sonication power, etc., are used to break the large particle?
Comments on the Quality of English LanguageNA
Author Response
Dear Reviewer,
We greatly appreciate the valuable comments and suggestions from the reviewer. We have carefully read and responded to all comments, point by point. The specific alterations in the manuscript in response to the reviewer’s comments are shown in yellow highlights. In addition, we have corrected English grammar in all the sections in the manuscript.
We hope all of the changes have addressed the reviewers’ concerns, so with these additions, we hope our work will be accepted for publication in Pharmaceutics.
Best regards,
Asst. Prof. Dr. Kanokwan Kiattisin

Reviewer 2 Report
Comments and Suggestions for Authors
This research aimed to evaluate the antioxidant, anti-inflammatory, and antibacterial activities of ethanol and ethyl acetate C. chinensis extracts. The results showed that both C. chinensis extracts exhibited antioxidant activity when tested by DPPH, FRAP, and lipid peroxidation inhibition assays. C. chinensis extract-loaded nanoemulsion-based gel had a positive effect on wound healing, presenting a percent wound contraction better than fucidin cream. However, there are some problems need to be addressed before publication.
1. At present, there are many studies using exosomes or cell membrane technology to maintain the original active ingredients while improving the delivery efficiency of drugs, such as Nature Communications, 2024, 15, 1042; Exploration, 2024, 20230164; Advanced Science, 2024, 2310211; Biomaterials, 2024, 306, 122478. In this study, the cell lysates were used to recommend the authors to analyze and compare the advantages and disadvantages of the two methods.
2. There are some in vitro cell experiments that are missing, such as anti-inflammatory assays, macrophage polarization, cell migration, etc., please refer to Nature Communications, 2024, 15, 1042; Materials Horizons, 2023, 10, 5474–5483
3. Animal experiments showed that DE-NEs-based gel did not significantly promote wound healing compared with the control group. We ask the authors to analyze the reasons and propose possible strategies to improve its efficacy.
4. In vivo experiments in animals also lack a large number of confirmatory data, please refer to Nature Communications, 2024, 15, 1042.
Comments on the Quality of English LanguageModerate editing of English language required
Author Response
Dear Reviewer,
We greatly appreciate the valuable comments and suggestions from the reviewer. We have carefully read and responded to all comments, point by point. The specific alterations in the manuscript in response to the reviewer’s comments are shown in blue highlights. In addition, we have corrected English grammar in all the sections in the manuscript.
We hope all of the changes have addressed the reviewers’ concerns, so with these additions, we hope our work will be accepted for publication in Pharmaceutics.
Best regards,
Asst. Prof. Dr. Kanokwan Kiattisin

Reviewer 3 Report
Comments and Suggestions for Authors
pharmaceutics-2965785
Exploring the Wound Healing Potential of Cuscuta chinensis Extract Loaded Nanoemulsions-Based Gel
The manuscript by Nitthikan et al. described the development and evaluation of Cuscuta chinensis extract loaded nanoemulsions-based gel for wound healing. The manuscript was appropriately prepared and the data were, in part, sufficient for the conclusion. However, there are several issues to consider as follows.
1. Introduction: The research gap needed to be clarified. The rationale of this work should be clarified. What is the reason for encapsulating Cuscuta chinensis extract into nanoemulsions? What properties of the extract need to be improved by nanoemulsion?
2. Chemical Markers Analysis by High Performance Liquid Chromatography (HPLC): The authors should develop calibration curves of kaempferol and hyperoside to determine their concentration in DEA and DE. Using only a standard concentration may not be accurate.
3. What is Trolox in Table 3? It was not mentioned elsewhere in the manuscript.
4. Please provide original TEMs, which include the scale bars.
5. Wound healing effects: Data shown in Figures 9 and 10 indicate that the wound healing effects of the DE-NEs-based gel are slightly better than 0.9% NaCl and 2% Fucidin cream. Interestingly, 0.9% NaCl and 2% Fucidin cream show similar effects. The authors should discuss these data. Can we conclude that the DE-NEs-based gel is beneficial? Why did the authors not include the DE as a control to emphasize the necessity of developing DE nanoemulsions? In the end, is the development of nanoemulsions and nanoemulsion-based gels necessary?
6. Lines 79 – 84: Please include relevant references.
7. What is “VP” in “ammonium acryloyldimethyltaurate/VP copolymer” (line 363)?
Comments on the Quality of English Language
Minor editing of English language required
Author Response
Dear Reviewer,
We greatly appreciate the valuable comments and suggestions from the reviewer. We have carefully read and responded to all comments, point by point. The specific alterations in the manuscript in response to the reviewer’s comments are shown in green highlights. In addition, we have corrected English grammar in all the sections in the manuscript.
We hope all of the changes have addressed the reviewers’ concerns, so with these additions, we hope our work will be accepted for publication in Pharmaceutics.
Best regards,
Asst. Prof. Dr. Kanokwan Kiattisin

Round 2
Reviewer 1 Report
Comments and Suggestions for Authors
Acceptable in the revised form.
Reviewer 2 Report
Comments and Suggestions for Authors
Accept.
Reviewer 3 Report
Comments and Suggestions for Authors
The revised manuscript can be accepted as is.